# Using Drones for Thermal Imaging Photography and Building 3D Images to Analyze the Defects of Solar Modules

Kuo-Chien Liao [1,*] , Hom-Yu Wu [2,*] and Hung-Ta Wen [1]

1   Department of Aeronautical Engineering, Chaoyang University of Technology, Taichung 413, Taiwan; wenhungta@gmail.com
2   Department of Mechanical Engineering, Lungwa University of Science and Technology, Taoyuan 333, Taiwan
*   Correspondence: james19831111@gmail.com (K.-C.L.); wuhomyu@gmail.com (H.-Y.W.);
    Tel.: +886-982-365-503 (K.-C.L.); +886-919-962-552 (H.-Y.W.)

**Abstract:** In this research, drones were used to capture thermal images and detect different types of failure of solar modules, and MATLAB® image analysis was also conducted to evaluate the health of the solar modules. The processes included image acquisition and transmission by drone, grayscale conversion, filtering, 3D image construction, and analysis. The analyzed targets were the solar modules installed on buildings. The results showed that the employment of drones to monitor solar module farms could significantly improve inspection efficiency. Moreover, by combining the mean and median filtering techniques, an innovative box filtering method was successfully created. Additionally, this study compared the differences between the mean, median, and box filtering techniques, and proved that the 3D image improved by box filtering is a more convenient and accurate way to check the health of solar modules than the mean and median filtering methods. In addition, this new method can simplify the maintenance process, as it helps maintenance personnel to determine whether to replace the solar modules on site, achieving the goal of power generation efficiency enhancement. It is worth noting that 3D image recognition technology can enhance the clarity of thermal images, thereby providing maintenance personnel with better defect diagnosis capability. It is also able to provide the temperature value of the defect zone, and to indicate the scale of defects through the cumulative temperature chart, so the 3D image is qualified as a quantitative and qualitative indicator. The analysis of the transmitted image is innovative that it not only can locate the defect area of the module, but also can display the temperature of the module, providing more information for maintenance personnel.

**Keywords:** drone application; image processing; IR image analysis

## 1. Introduction

The Sustainable Development Goals (SDGs) and global climate change are important global issues today. There is a mandate for all countries to cooperate and take practical and appropriate transnational actions to rapidly reduce greenhouse gas emissions on a global scale [1]. Against this backdrop, further research and extensive application of green energy are all the more important and critical. The purpose of this research is to develop an advanced solar module monitoring and analysis system that can effectively analyze the health status of solar modules and provide maintenance personnel with accurate information for regular maintenance, aiming to extend the service lifespan of solar modules.

Currently, "power generation efficiency" is the parameter to monitor the performance of both large and small solar photovoltaic plants. This monitoring method fails to satisfy the daily and preventive maintenance requirements to extend the service lifespan of photovoltaic systems. In addition, many solar modules are installed on buildings and vast areas such as lakes. It is a considerable challenge to carry out maintenance and monitoring of solar modules in such locations.

A solar plant is composed of a series of connected modules. The shadow on one solar cell affects the whole series, thereby reducing the power generation efficiency of the entire solar system. However, module shadow can be caused by many factors. One such factor is the onsite environmental impact of things such as forests, nearby buildings, clouds, sand and dust accumulation, bird droppings, etc. The power generation efficiency of solar modules and the energy reduction caused by partial shadow are counted at 5–25% a year [2]. This is mainly because the shadow increases the internal impedance of the solar cell and blocks the current path, and then a reverse-current situation occurs. Under this situation, the blocked current is converted into heat loss, which makes the solar cell heat up locally and, finally, causes hot spots, yellowing, and glass breakage [2].

As mentioned above, the temperature of the solar module is an essential factor to demonstrate power generation efficiency. For instance, when the temperature rises by 1 °C, the power generation efficiency of the solar cell module with the defect decreases by 0.5% [3–6]. Alsafasfeh et al. proposed a safer and low-cost real-time model combining two cameras, a thermal imager, and a charge-coupled device (CCD) mounted on a drone, to indicate and detect the faults in a PV system [4]. According to the experiment, it could detect internal and external faults. Rosell and Ibanez proposed a methodology to estimate PV electrical production by modifying the I–V model curve. This adjustment could create a new maximum power output expression and provide PV module performance parameters for all operating conditions [5]. Hwang et al. presented a method to analyze the defects of PV systems by using modules' temperature, power output, and panel images. This method is able to check for failures rapidly, such as hot spots, panel breakage, connector breakage, busbar breakage, panel cell overheating, and diode failure [6].

Many scientists have been endeavoring to develop an accurate and effective method to monitor and identify solar module failures [7,8]. Yahyaoui and Segatto proposed a technology to monitor and detect the faults of a single-phase grid-connected PV plant. This method used two current and voltage indicators to analyze the faults affected by bypassed PV modules, open-circuit strings, and partial shading to discover the total number of faulty PV modules and strings [7].

Liao and Lu employed an unmanned aerial vehicle (UAV) to conduct the detection of solar panel faults by inspecting solar panel infrared (IR) images. These infrared image displays could be divided into three health conditions using the MATLAB® image analysis toolbox [8].

As timely detection of solar module failures can prolong the service lifespan and maintain the solar system's performance [8,9], Madeti and Singh proposed a review of a monitoring system for photovoltaic plants. The sensors, the controller used in the data collection systems, and their working principles, as well as data transmission, data storage, and analysis methodologies, are to be examined in this paper, with the aim to build an effective, low-cost, and viable PV monitoring system that can reach our desired performance [9].

It is also important to improve the safety of maintenance personnel when performing maintenance duties in high-risk areas such as high-rise buildings and lakes. Jeong proposed an effective matching method for feature points and a homography translation technique. The temperature data derivation method and the standard/abnormal decision method were adopted to enhance the performance [10]. However, solar defect module analysis was not performed in Jeong's research; our proposed method (improved box filtering) can fill in the missing pieces.

Navid et al. and their proposed methodologies were compared and validated by our team using thermal imaging. The results showed that the proposed fault monitoring scheme can effectively monitor and characterize faults compared with traditional thermal imaging [11–17]. Therefore, our study adopted UAVs installed with a thermal imager as the acquisition tool for solar module images.

Ballestín-Fuertes attached mechanical components to the current solar PV modules to enhance the automation of monitoring. In this study, the add-on components and their

configuration were investigated, and a control strategy for applying this technology to large photovoltaic plants was developed. A PV inverter was designed to validate the proposed methodology on a small-scale solar plant to obtain onsite EL images of an actual plant [18]. It was found that the cost of this method is high, and is not applicable for large-area monitoring.

Sciuto et al. proposed an elliptical neural network method with feature extraction based on the co-occurrence matrix and SVD decomposition. This technology effectively detects different types of organic solar cell (OSC) surface defects, such as cracks, breaks, and scratches. Its classification accuracy is up to 95.4% [19]. Moreover, in 2021, Sciuto et al. proposed a new method to possibly detect and classify the different kinds of defects occurring in OSCs' manufacturing process. The Zernike moments were used to extract the features from the scanning electron microscope images. These features were then used as the input for the feedforward probabilistic neural network model (EBFNN). After training, the correct classification of the model was 89.3% over the testing dataset [20].

As it is urgent to develop viable and sustainable green energy, the maintenance of solar modules and monitoring of their conditions has become an important issue recently. Many studies have been conducted to find a quick and reliable method of fault detection. A review of the recently published papers is listed in Table 1, illustrating how these investigations were conducted. This clearly indicates that UAVs are a convenient tool for large-scale monitoring. Many researchers use drones for collecting data, and IR image detection is widely adopted, with many researchers using it for fault detection.

**Table 1.** A comparison of references related to solar panel monitoring using drones.

| References | Method | |
| --- | --- | --- |
| | **Drones + IR Image Defect Detection** | **Post-Detection Analysis** |
| Alsafasfeh, M., et al. [4] | Yes | |
| Jeong, H., et al. [10] | Yes | Diagnosis |
| Navid, Q., et al. [11] | Yes | |
| Henry, C., et al. [12] | Yes | |
| Pierdicca, R., et al. [13] | Yes | Deep learning |
| Boulhidja, S., et al. [14] | Yes | |
| Tsanakas, J.A., et al. [15] | Yes | |
| Gallardo-Saavedra, S., et al. [16] | Yes | |
| Herraiz, Á.H., et al. [17] | Yes | |
| Ballestín-Fuertes, J., et al. [18] | | EL [1] |
| Zhang, H., et al. [21] | | Stacking model |
| López Gómez, J., et al. [22] | | Artificial neural networks |
| Ponce-Jara, M.A., et al. [23] | | IoT monitoring system |

[1] Electroluminescence technique.

Utilizing two PV datasets, Zhang developed four different stacking models—based on extreme gradient boosting, random forests, light gradient boosting, and gradient boosting decision trees—to predict photovoltaic power generation [21].

López Gómez fed GDAS weather data into an ANN model; the tested numerical weather model could be combined with machine learning tools to model the output of PV systems with less than 10% error, even when in situ weather measurements were not available [22]. The above two scholars were able to predict the defects by estimating the power generation, although the defective areas could not be precisely located.

Ponce-Jara proposed that a photovoltaic system connected to an IoT monitoring system with dual-axis tracking produces 19.62% more energy than a static photovoltaic system [23]. The above research can improve energy efficiency, but it neglects the aspect of physical defect monitoring. It is suggested that the method proposed by Ponce-Jara can be used for back-end analysis to determine the real-time defect status of PV modules.

Table 2 summarizes other researchers' relevant methods and their key points in solar module monitoring.

**Table 2.** Key points of each study in the literature review.

| No. | Authors | Reference | Key Point |
| --- | --- | --- | --- |
| 1 | Jeong, H., et al. | [10] | Using the maximally stable extremal regions (MSER) method, which proposes an effective matching method for feature points and a homography translation technique. The derivation method and the normal/abnormal decision method are described. |
| 2 | Pierdicca, R., et al. | [13] | Intersection over union (IoU) is trained and evaluated on the photovoltaic thermal image dataset—a publicly available dataset collected for this work. |
| 3 | Ballestín-Fuertes, J., et al. | [18] | Demonstrates the technical feasibility of onsite EL inspection of photovoltaic power plants without measuring and analyzing panel defects of photovoltaic installations. |
| 4 | Zhang, H., et al. | [21] | Using two PV datasets for gradient boosting, random forests, light gradient boosting, and gradient boosting decision trees to predict photovoltaic power generation. |
| 5 | López Gómez, J., et al. | [22] | Feeds GDAS weather data into an ANN model; the tested numerical weather model can be combined with machine learning tools to model the output of PV systems. |
| 6 | Ponce-Jara, M.A., et al. | [23] | PV modules are connected to an IoT monitoring system with dual-axis tracking. |

Given the above, this study proposes an innovative approach utilizing an infrared imaging system with UAV functionality to detect faults in solar power systems; the acquired IR images were also reproduced to construct 3D images for analysis and comparison purposes. Furthermore, the defect location and relative temperature of the solar module were obtained by comparison with the normal operating module and the color bar of the 3D image. The defect locations were then analyzed using an IR image with an irradiance of 500 W/m$^2$ instead of above 700 W/m$^2$, which was the parameter proposed by previous studies for IR imaging. Our proposed new method is obviously a breakthrough, as it can be applied to obtain the best IR image even at an irradiance of 500 W/m$^2$.

## 2. Detection Method and Experimental Setup

### 2.1. Visual Inspection

Visual inspection is regarded as one of the key inspection methods. Nevertheless, solar modules are usually installed on building rooftops or in vast areas, where it is not easy for maintenance personnel to perform inspections. Utilizing flexible and maneuverable drones equipped with visible light sensors to perform inspection tasks can effectively increase inspection efficiency and save manpower. Despite this, inspection of visible-light images can only be used to inspect the appearance of defects, and cannot further investigate the interior of the solar cells. Hence, electroluminescence technology (EL) or thermal image evaluation is required for in-depth examination.

### 2.2. Electroluminescence (EL)

Electroluminescence (EL) technology makes use of electroluminescence images to show material defects, microcracks, process contamination, sintering waves, and broken fingers of solar modules [24,25].

The mechanism is to add a forward current to the solar panel to make it become a light-emitting diode. If there exists cracks or flaws, defect voids will be generated, and light will be emitted. The light intensity is proportional to the input current, and can

reflect the density of defects; the part with fewer defects has a stronger luminous intensity, whereas the part with more defects has a relatively weak luminous intensity. By means of electroluminescence images, defects such as material defects, microcracks, and circuit breakages in solar cells can be clearly identified.

Although electroluminescence technology can help diagnose faults or defects, this technology can only detect one solar module at a time, and the EL equipment has to be placed close to the solar module to perform the inspection. Therefore, this method is considered to be labor-intensive and time-consuming. Therefore, the application of electroluminescence technology is mainly confined to laboratories.

### 2.3. Thermal Imaging Inspection (IR)

Thermal imaging inspection is a novel method that provides data about the condition of solar modules. Usually, it is performed using a sensor with thermal images (IR images). Thermal image evaluation is a harmless and non-contact monitoring technology that can directly diagnose the defects and failures of most solar modules, but this inspection method is applicable only under appropriate ambient temperature, wind speed, and sunlight (at least 700 W/m$^2$ radiation) [26,27]. Despite this, this method is still valid, as it can inspect the interior conditions of solar cell modules. In addition, hot spots (defects) caused by abnormally high temperatures can be detected by comparing the visible and infrared images taken at the exact location of the solar photovoltaic farm. One noteworthy point is that IR detection produces noise, so filtering is needed to reform the image features and improve defect recognition ability. Furthermore, if the shooting angle is not directly facing the solar module, misjudgment will occur.

### 2.4. Comparison of Detection Methods

The typical solar module testing methods are described in Table 3. In this study, analyses and comparison were carried out to devise a fast, time-saving, and low-cost method for solar system inspection. It is shown in Table 3 that the drone + IR image method can meet all of our criteria. As mentioned earlier, IR detection produces noise, and the shooting angle must directly face the solar module in order to prevent misjudgment and other problems. In this research, drones integrated with IR sensors were evaluated thoroughly. Our ultimate goal is to develop a real-time detection system that is efficient, time-saving, and safe, and at the same time can be used sustainably to monitor and identify defects in solar modules.

**Table 3.** Comparison of solar module detection methods and purposes.

| Detection Method | Inspection Purpose and Requirements | | | | |
| --- | --- | --- | --- | --- | --- |
| | Cell Surface Inspection | Cell Internal Inspection | Fast Save Time | Safety | Outdoor |
| Visual inspection | Yes | | | Yes | |
| EL [1] | | Yes | | | Yes |
| IR image [2] | Yes | Yes | | Yes | Yes |
| Drone + IR image | Yes | Yes | Yes | Yes | Yes |

[1] Electroluminescence; [2] thermal imaging inspection. The table is derived from [3,4,24–29].

### 2.5. Typical Filtering Techniques Introduction

Image filtering is a technique to eliminate the image's noise while keeping the image's main features as much as possible. This is essential in the execution of image preprocessing. Given the fact that the quality of the processing result directly affects the validity and reliability of image processing and analysis, different types of filtering techniques—such as median filtering and mean filtering—are introduced below.

### 2.5.1. Median Filtering

This filtering method is a nonlinear signal processing technology that can effectively suppress noise using the statistical sorting theory [30]. The primary filtering principle is to select the median value from the digital image or digital sequence and then adopt the median value. As the discounted value replaces its neighboring value to draw the surrounding pixel values closer to the actual value, independent noise points are then successfully eliminated.

Median filtering can remove intense high-frequency noise while still maintaining the sharpness of the edges. Nevertheless, this filtering method is not suitable for removing significant area noise, and can best tackle spot noise only, such as salt and pepper noise. The method uses a two-dimensional matrix to sort the pixels in the matrix according to the size of the pixel values to form a two-dimensional data sequence, as shown in Equation (1):

$$\sum_{i=1}^{N} |x_{med} - x_i| \leq \sum_{i=1}^{N} |x_j - x_i| \text{ for } j = 1 \ldots N \tag{1}$$

Figure 1 is an example of filtering using a 3 × 3 two-dimensional matrix. After reordering the pixel values in the 3 × 3 matrix—10, 15, 20, 20, 20, 20, 20, 25, 100—the median value is 20, and the middle value and the gray of the pixels in the matrix can be defined. Finally, the order value is replaced by the intermediate value. It can be seen that the median filter can remove high-frequency solid noise while still maintaining the sharpness of the edge.

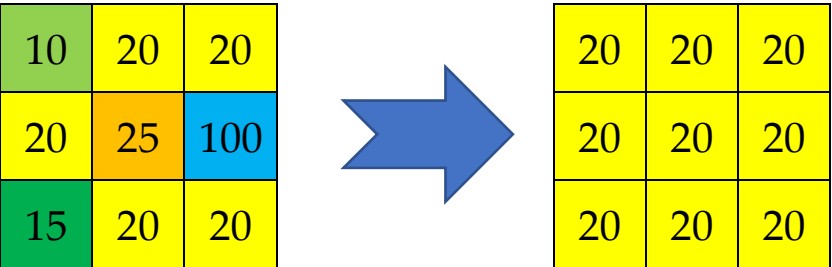

**Figure 1.** Median filtering model (3 × 3 matrix).

### 2.5.2. Mean Filtering

The averaging filter converts the image into pixel values (0–255), and then adds the pixel values in the matrix and averages them, as shown in Equation (2):

$$g(x,y) = \frac{1}{m} f(x,y) \tag{2}$$

m: Pixels in the matrix.

The mean filter is also a low-pass filter. Having averaged the values in the matrix, the values in the original matrix are replaced. As shown in Figure 2 (3 × 3 matrix), the matrix of the mean filter is averaged by standard pixels. The primary purpose is to use fuzzy pictures to deduce a rough description of the beautified image, in which irrelevant noise in the image is eliminated.

### 2.6. Experimental Setup and Innovative Methods

The experimental setup used UASs (unmanned aircraft systems) equipped with thermal image sensors. This is a more economical solar module monitoring system, with fast detection and efficient analysis capabilities. The drone flew over the solar photovoltaic farm and took photos of the solar modules, and then the images were sent back to the ground control station for analysis.

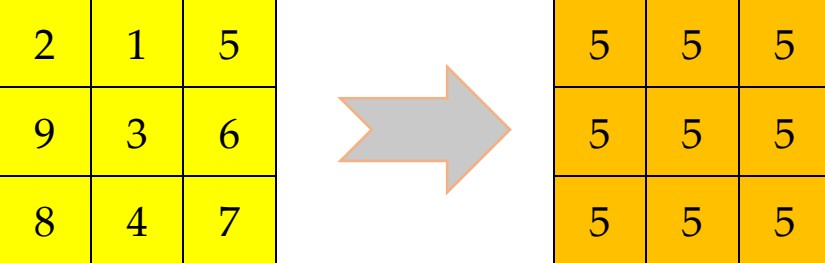

**Figure 2.** Mean filtering model (3 × 3 matrix).

### 2.6.1. Experimental Setup

The drone model was DJI-Mavic 2 Enterprise (Da-Jiang Innovations, Shenzhen, China), as shown in Figure 3a. This is a light and portable drone that can be assembled quickly, so that the setting time of the ground control station can be shortened to facilitate rapid detection. Functioning as a planned system for monitoring solar modules, FLIR Lepton® thermal image microsensors were also integrated (shown in Table 4) for capturing visible-light and IR images. The inspection procedures were as follows: (1) drones were equipped with sensors to fly over the solar module and take aerial images, including visible-light images and IR images, and take temperature records of the solar module; (2) images were transmitted to the ground station for image analysis through the radio frequency (RF) channel; (3) the ground station recorded the module's images and conducted real-time analysis of the module's health status. However, the image quality may be affected by many factors; for example, weather conditions, irradiance, shadow, or sunlight reflection during the inspection may cause the quality of infrared images to deteriorate, leading to failure in defect identification. Therefore, practical analysis tools should also be applied to facilitate defect and failure detection.

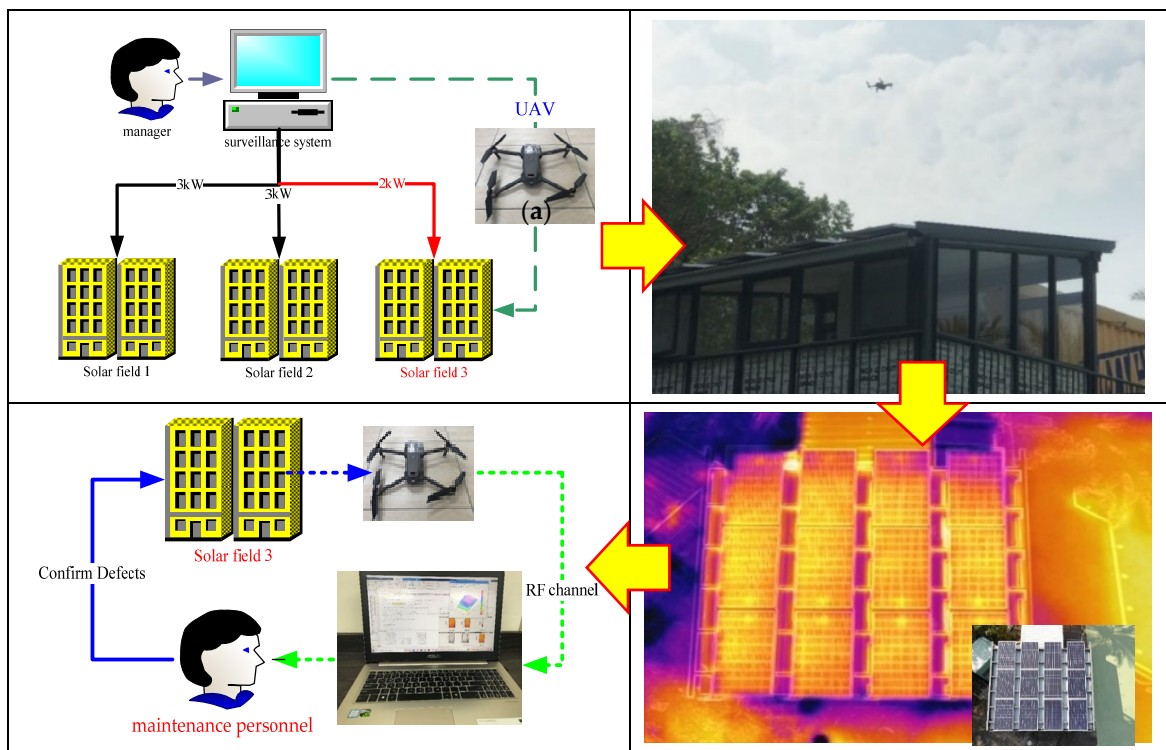

**Figure 3.** Flight mission planning—CYUT solar photovoltaic experimental field.

**Table 4.** FLIR Lepton® specifications.

| No. | LEPTON 3.5 | Specification |
|---|---|---|
| 1 | Effective frame rate | 8.7 Hz |
| 2 | Output format | 14-bit, 8-bit, 24-bit RGB |
| 3 | Pixel size | 12 μm |
| 4 | Scene dynamic range | Low-gain mode: −10 to 400 °C; High-gain mode: −10 to 140 °C |
| 5 | Spectral range | 8 μm to 14 μm |
| 6 | Thermal sensitivity | <50 mK (0.050 °C) |
| 7 | Visual angle | 57 |
| 8 | Resolution | 160 × 120 |

It is generally known that the defects of solar modules may be caused by different physical properties, leading to the malfunction or reduced efficiency of the modules. Physical defects such as snail trails, shadows, hot spots, microcracks, etc., can lead to the increase in the cell's temperature and the formation of hot spots, or even malfunction of the cell. Therefore, this study focuses on analyzing solar modules' surface temperature changes. With the help of these temperature changes, the zone and the exact area of possible defects can be indicated, thereby providing valuable information and references for maintenance personnel (as shown in Figure 3).

2.6.2. Innovative Methods

More than 40 years ago, J.-S. Lee, V. Frost, etc., proposed different principles to eliminate noise filtering [31–38]. Today, there has been prominent progress and achievements in noise elimination technology, but there is still no common opinion on the best filter and its related parameters. Therefore, a new filtering method—improved box filtering—is proposed for IR images in this research. This improved box filter is different from the traditional box filter [39]. It combines the advantages of both the median and mean filters. In addition to highlighting module defects, it can also suppress noise generation. The normal/defect areas in the module can thus be clearly displayed.

This is an innovative method for IR image analysis for solar modules. The use of median filtering in the MATLAB® (The MathWorks, Natick, MA, USA) environment is able to retain characteristic edges, and mean filtering is able to eliminate noise and edit related filtering parameters. Hence, box filtering not only can retain the distinctive edges, but also can achieve better filtering and smoothing effects, as shown in Equation (3):

$$b(x,y) = \frac{1}{n} f(x,y) \tag{3}$$

n: Pixels around the center of the matrix;
b: Matrix center pixel.

In addition, the improved box filter can be presented in a matrix *(hsize.width* hsize.height)*, as shown in Figure 4.

$$h = \frac{1}{hsize.width * hsize.height} \begin{bmatrix} 1 & 1 & 1 & ... & 1 & 1 \\ 1 & 1 & 1 & ... & 1 & 1 \\ ... & ... & ... & ... & ... & ... \\ 1 & 1 & 1 & ... & 1 & 1 \end{bmatrix}$$

**Figure 4.** Box filtering model.

Based on the abovementioned experimental configuration and image analysis, this research used drones to take images of solar modules and send them back to the ground station for immediate image analysis [38]. The analysis procedures are indicated in Figure 5.

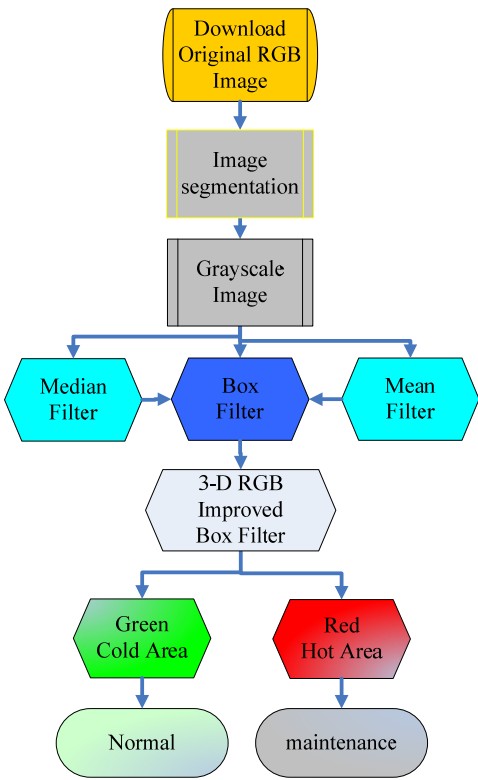

**Figure 5.** Flowchart of an image filtering process.

The flowchart shows the process of solar module monitoring. The first step in the algorithm is to convert the original color image to a grayscale image so that the final output image is more evident than the original image for easy identification of the defect location.

Following the primary analysis guidelines, the second step is to determine the brightness of the solar module surface. The clarity of brightness helps to distinguish defects, and the grayscale image can facilitate the identification of the defect location better than the original color image. By converting the grayscale and filtered images into a 3D image, it is a lot easier to identify the defect location. Therefore, the filtering is a critical step in order to achieve our desired results.

In the third step, the mean and median filters are utilized to improve the block filter. Beforehand, image analysis has to be performed to confirm the suitability of various filters.

Finally, the best analysis method is selected and processed. The post-analysis images are then provided for maintenance personnel as a reference.

## 3. Results

The identification of the solar cell defect depends upon the comparison of the deviation of the cell surface temperature. It is also related to the ambient temperature, wind speed, and weather conditions. It is known that at least 700 W/m$^2$ of irradiance is required for the solar module inspection. Moreover, there are a series of external factors that affect the solar cell's working environment temperature, such as sunlight intensity, ambient temperature, particle radiation, etc. These all are performance indicators of the solar cell. This study compares the normal and defective modules with different sunlight intensities and ambient temperatures. Firstly, the experiment was conducted from 8 a.m. to 1 p.m. on 9 July 2021. The outdoor temperature was 27–32 °C, and the wind speed was 1 m/s. The irradiance during the experiment was about 500–850 W/m$^2$. This experiment was carried out under

the same sunlight intensity and ambient temperature, with two solar modules selected for the investigation. One of them was a normal operating module, and the other had a prefabricated "shadow" defect, which was sprayed with waterproof paint on the surface of the cells. The shaded (yellow) fault was divided into two regional defect locations, as shown in Figure 6. There was a total of 10 defective solar cells (defect rate of about 16.7%). If the purple area affected by the junction box (which is 1.5%) was included, the total defect rate reached about 18.2%. The junction box, as shown in Figure 6c, is a relay station for solar modules to convert to electrical energy. Its primary function is to connect the electricity generated by the solar module to the external circuit. The junction box was fixed at the back position above/below the middle of the module, and it covered about 1.5% of the module area. Under normal operation, the thermal energy generated by the junction box affects the cell surface temperature. Therefore, the temperature of the junction box area (about 1.5% of the cell) is likely to be relatively higher, as shown in the grey frame in Figure 6b.

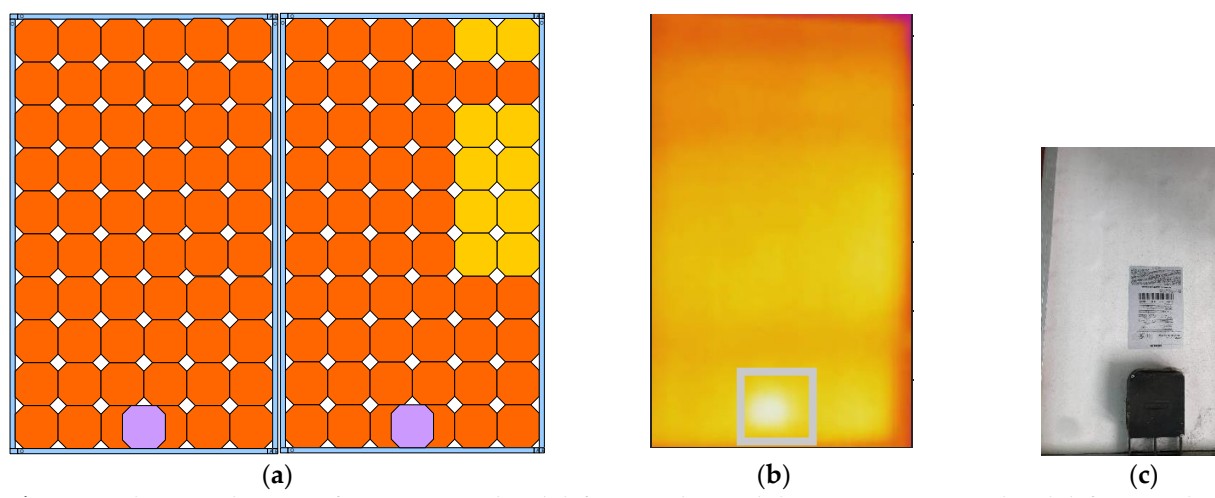

**Figure 6.** Schematic diagram of preset normal and defective solar modules: (**a**) Preset normal and defective solar modules. (**b**) Junction box for the IR image. (**c**) Junction box.

In this section, the differences between the box, median, and mean filters are compared. We found that defect locations cannot be accurately distinguished by median and mean filtering, but the scope of the defects can be easily identified by box filtering. Significantly, the contribution of this research was more significant at 500 W/m$^2$. This method can not only clearly determine the position of the defect, but also can obtain the relative temperature of the defect by using the analyzed color bar. In the following paragraphs, the box, median, and mean filtering methods for three different degrees of defects are described. In accordance with other studies, the irradiance of 700 W/m$^2$ was used for the experimental control group.

### 3.1. Irradiance at 700 W/m$^2$

Figure 7a shows the original IR image of the normal operating module (left side) and the defective preset module (right side, green frame). As it is difficult to identify the defect location and range in the original IR image, we used image analysis to strengthen defect recognition in this research. After processing with the mean filter and grayscale, it was found that the average filter could remove the noise caused by the IR image, as shown in Figure 7b. Figure 7c indicates that the image processed with median filtering can highlight the preset defects. At the same time, this study also developed a box filter for comparison, as shown in Figure 7d. The preset defects could then be identified under the irradiance of 700 W/m$^2$.

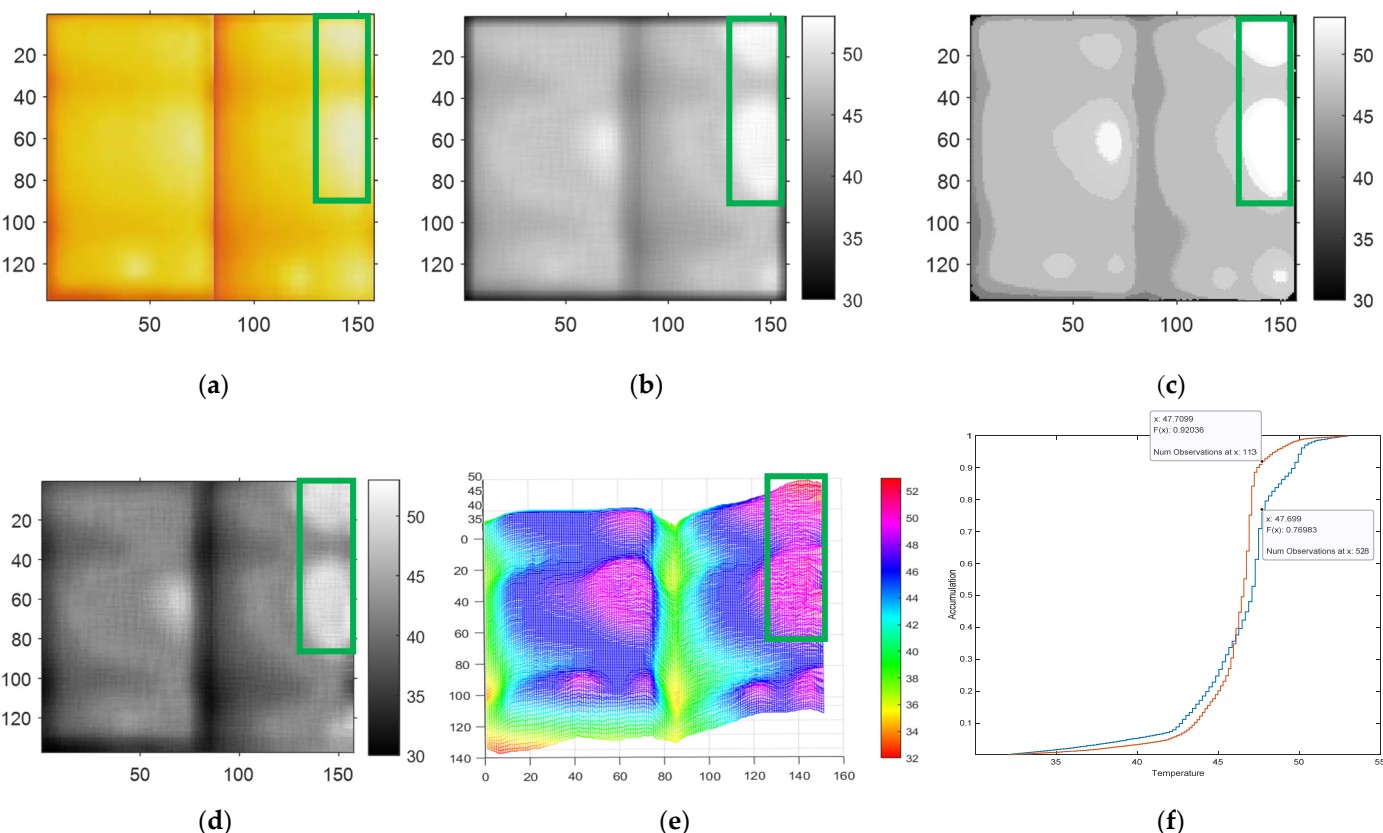

**Figure 7.** Comparison of different filtering methods with irradiance at 700 W/m²: (**a**) Original IR image. (**b**) Mean filter after grayscale treatment. (**c**) Median filter after grayscale treatment. (**d**) Improved box filter after grayscale treatment. (**e**) Improved box filter after 3D image. (**f**) Cumulative chart comparison of normal operation and abnormal conditions of solar modules.

Furthermore, the box-filtered image after grayscale treatment is presented in 3D (Figure 7e) to illustrate a better recognition effect. By comparing the temperature of the relative positions of the two modules, it can also be seen that the cell's face temperature of the defect position is about 2–3 °C higher than the temperature of the normal operating module. At the same time, this study converted the temperature of the IR image into a cumulative chart, and compared the temperature of the normal operating module and the defective module, as shown in Figure 7f. The red line in the accumulation graph is the temperature accumulation line of the normal operating module, and the blue line is the defective module. By comparing the two modules, we found that the temperature of the defective module was about 2–3 °C higher than that of the normal module. In the cumulative chart, there is a 16% temperature difference between the red and blue lines at 47.7 °C, which is consistent with the preset module defect area (approximately 10 cells, occupying 16.7% of the total area).

### 3.2. Irradiance at 500 W/m²

As shown in Figure 8a, it is difficult to identify the defect location and range in the original IR image under the irradiance of 500 W/m². It was found that the mean filtering after grayscale treatment could remove the noise caused by the IR image, but the preset defect area could not be fully revealed, as shown in Figure 8b. On the other hand, the image processed by median filtering could highlight preset defects, as shown in Figure 8c, but it also expanded the scope of defects, leading to misjudgment of defective areas. It is noteworthy that when the cell's temperature was not entirely converted into electrical energy, the temperature of the defective area was higher than that of the normal area.

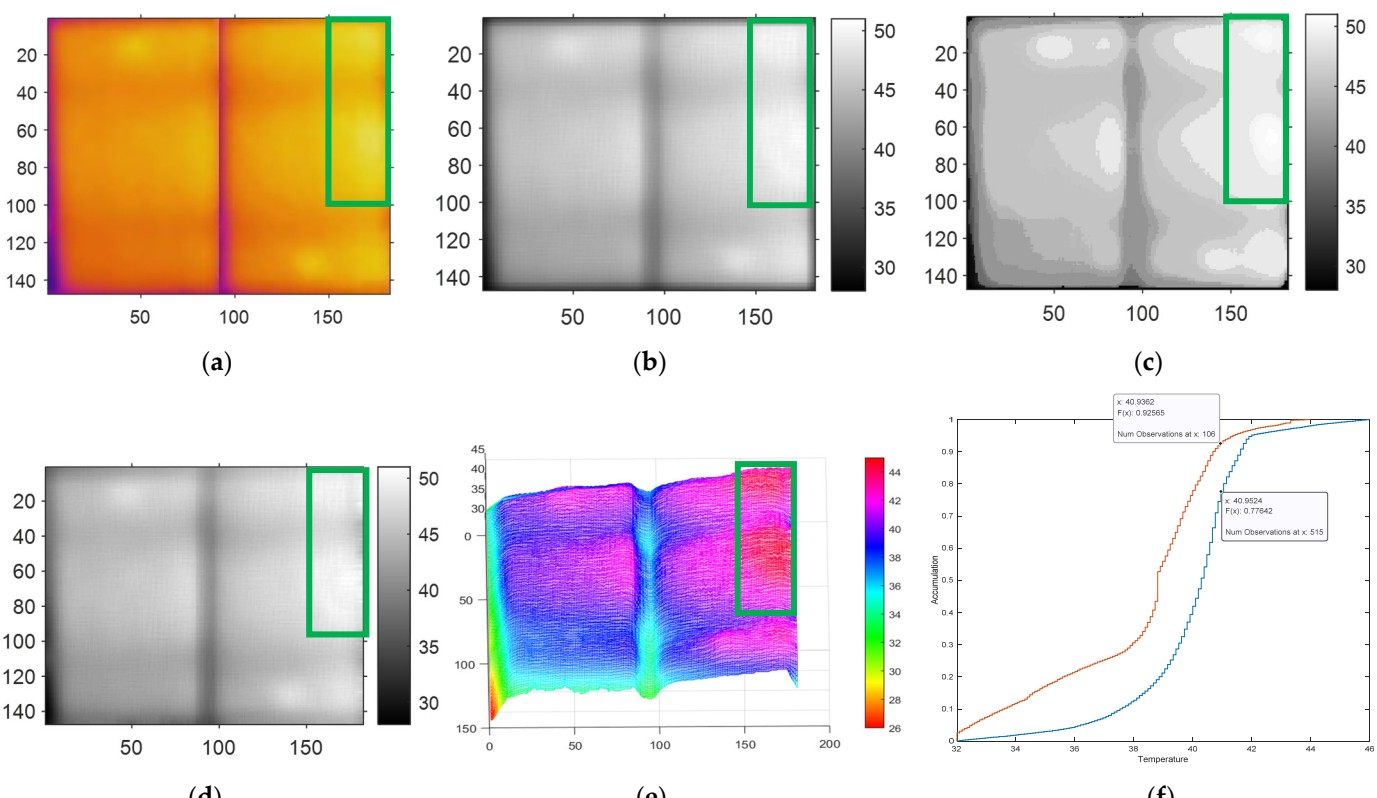

**Figure 8.** Comparison of different filtering methods with irradiance at 500 W/m². (**a**) Original IR image. (**b**) Mean filter after grayscale treatment. (**c**) Median filter after grayscale treatment. (**d**) Improved box filter after grayscale treatment. (**e**) Improved box filter after 3D image. (**f**) Cumulative chart comparison of normal operation and abnormal conditions of solar modules.

By the same token, when the sunlight is insufficient, the temperature difference between the defective and normal areas is less obvious. Therefore, one of the reasons for the expansion of the defective area is that there is no significant temperature difference between the faulty and normal-functioning cells. The box filtering technology developed in this research is advantageous in that it combines the median filter's retention of edge characteristics and the mean filter's effective noise elimination capabilities. With the application of box filtering technology, the analysis results of the defect range are closer to the actual preset defect range, as shown in Figure 8d.

Furthermore, the box-filtered image after grayscale treatment is presented in 3D to generate a better defect recognition effect. Under the irradiance of 500 W/m², the preset defects were fully apparent and the temperature was 45 °C (red zone), as shown in Figure 8e. After conducting the temperature comparison, it can also be seen that the surface temperature of the cell at the defective location is about 2–3 °C higher than the temperature at the normal location.

In the module comparison, as shown in Figure 8f, it can also be seen that the temperature of the defective modules is about 2–3 °C higher than that of the normal modules. In the cumulative chart, there is about 15% temperature difference between the red and blue lines at 40.9 °C, which is also close to the 16.7% module defect area (approximately 10 cells).

### 3.3. Irradiance at 850 W/m²

In this part, the irradiance of 150 W/m² higher than that of the control group was adopted for comparison. When the irradiance was 850 W/m², it was not easy to identify the defect location and range in the original IR image, as shown in Figure 9a.

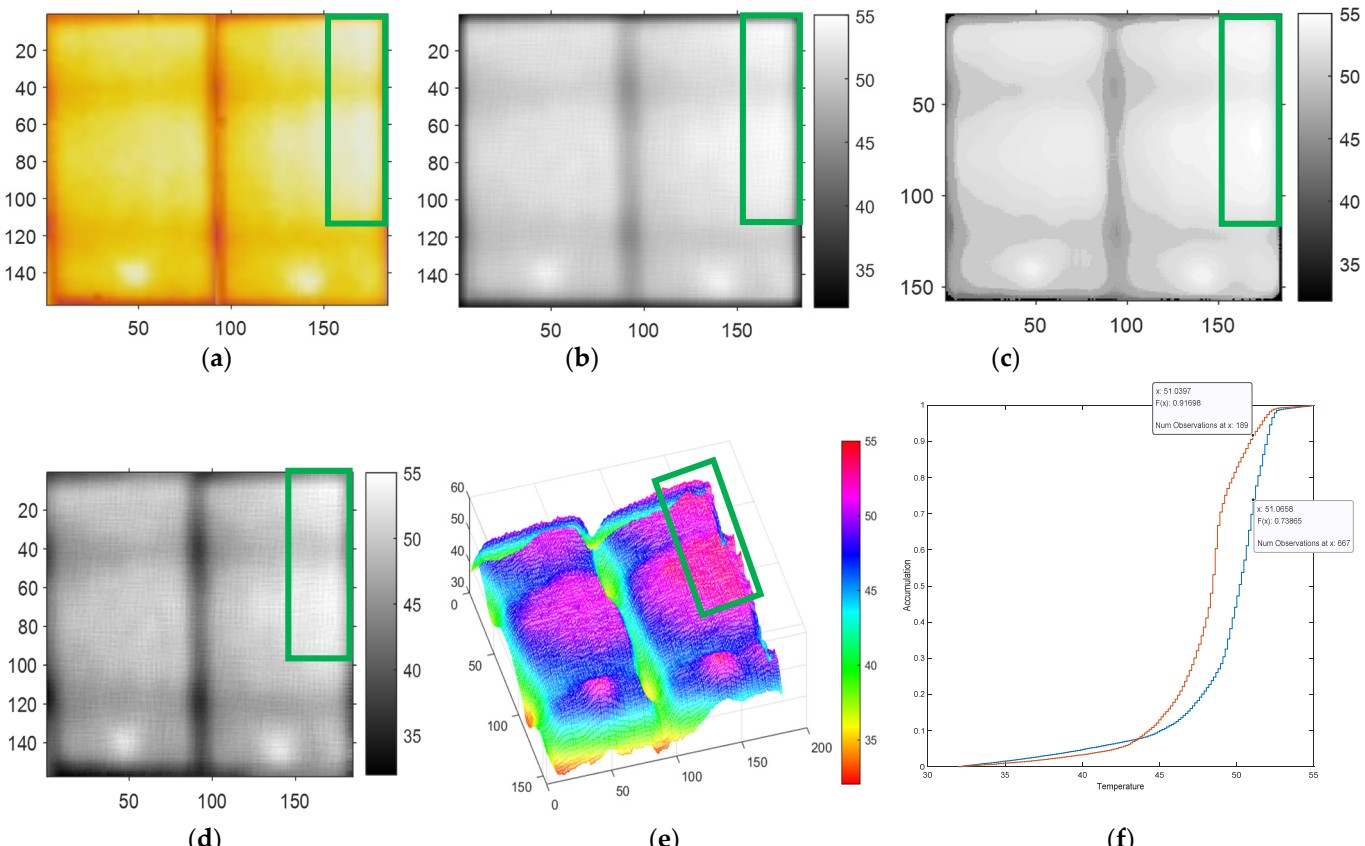

**Figure 9.** Comparison of different filtering methods with irradiance at 850 W/m$^2$: (**a**) Original IR image. (**b**) Mean filter after grayscale treatment. (**c**) Median filter after grayscale treatment. (**d**) Improved box filter after grayscale treatment. (**e**) Improved box filter after 3D image. (**f**) Cumulative chart comparison of normal operation and abnormal conditions of solar modules.

The mean filter after grayscale treatment was used to remove the noise caused by the IR image, and still the IR image could not fully reveal the preset defect area, as shown in Figure 9b. The image after median filtering also could not fully indicate the preset defects, as shown in Figure 9c. Such conditions may lead to a misjudgment of defective areas. After using the improved box filter analysis, the defect range, which is the red zone of the green frame shown in Figure 9e, could be seen. At the same time, the result was closer to the actual preset defect range, as shown in Figure 9d.

In the module comparison as shown in Figure 9f, it can also be seen that the temperature of the defective modules is about 2–3 °C higher than that of the normal modules. In the cumulative chart, there is about 18% temperature difference between the red and blue lines at 51 °C, which is also consistent with the 16.7% module defect area (approximately 10 cells). At the same time, this study found that the higher the irradiance intensity, the larger the defect area. Conversely, when the irradiance was lower (e.g., at 500 W/m$^2$), the defect area was reduced.

### 3.4. The Practical Verification Experiment

The experiment was conducted on panels with unknown defects. The location of these panels was at the GPS coordinates of 24°12′31″ N and 120°29′40″ E, with an altitude of 1 m. The experiment was carried out at 10 a.m. on June 3, 2022. The outdoor temperature was 30 °C, and the irradiance was 515 W/m$^2$.

There was a defective cell in the lower left of the panel. Figure 10a shows the original IR image, with a clear and bright area in the lower left (green frame) that corresponds to

the defected cell, and a less bright area located in the upper middle (the junction box area, as mentioned in Section 3 (grey frame)).

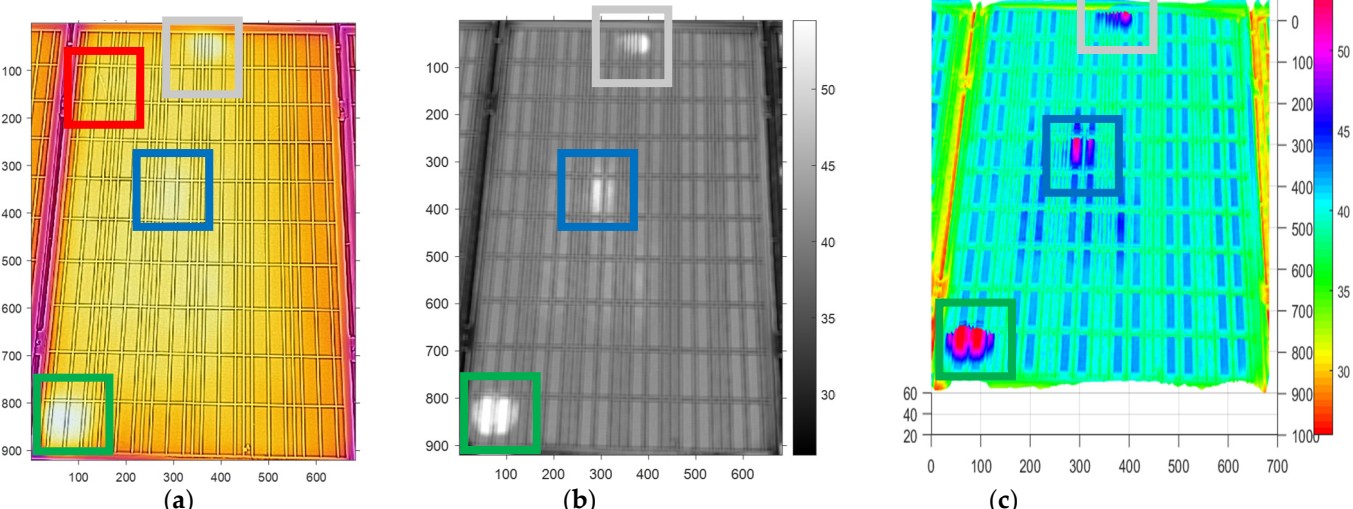

**Figure 10.** Verification experiment with irradiance at 515 W/m$^2$: (**a**) Original IR image. (**b**) Improved box filter after grayscale treatment. (**c**) Improved box filter after 3D image.

There was an inconspicuous crack or snail trail on the top left of the panel (red frame), while a less bright area located in the middle was the hot spot (blue frame). Figure 10a shows purple lines on the left and right margins, which are the frames of the panel. Figure 10b shows an image converted to grayscale after improved box filtering. The grid lines of the panel are blurred due to the filtering effect, but the bright region can still be seen clearly. Figure 10c displays a 3D color image after improved box filtering to differentiate temperatures, and the image shows that the hot spot is at about 54 °C, while the panel frames are at a temperature of 35 °C, and the rest of the panel is at around 40~45 °C.

Common causes of solar module failures include resistive or solder failures, cell failures, and cell interconnection defects. In the case of cell cracks and snail traces, they may cause an increase in temperature for part of the module, and gradually cause overall cell defects. At the same time, the overall temperature of the cell surface will increase or be higher than that of the adjacent cells. This study aimed to assess the resulting cell defects. During the process, method validation in different regions was carried out, and the results were as follows:

In Figure 10a, about 2–3 hot spots are shown on the solar module; the hot spot on the left bottom corner is visible (green frame). The hot spot within the grey frame is also easily seen, but that hot spot is caused by the junction box's rising temperature, and should not be considered a defect. The hot spot near the blue frame is not visible, but after treatment with greyscale (Figure 10b) and 3D image processing (Figure 10c), the defect location and scope can be clearly identified; finally, the three hot spots are clearly shown on the solar module in Figure 10c. Returning to Figure 10a, the crack on the upper left corner (red frame) of the module leads to a hot spot, but it does not yet cause a cell failure (cell hot spot), so no abnormal condition is detected. This experiment demonstrates that our new method involving the construction of 3D images can identify and analyze the defect zone when the irradiance is about 500 W/m$^2$. The most important contribution of this method is that it overcomes IR detection's obscurity problem.

## 4. Discussions

In this study, a more accurate monitoring method for checking the health of solar modules was successfully developed by integrating drones and IR cameras, along with an improved box filter to improve 3D image analysis technology. In addition, measurements

and comparisons were also carried out under different weather conditions, and the acquired images were also analyzed under varied sunlight conditions, with 500 W/m$^2$ or higher irradiance. As listed in Table 5, a comparison was made for different filtering methods. When the irradiance was 700 W/m$^2$, misjudgments occurred, as the defect scope was exaggerated as a result of the median filter. Otherwise, other filtering methods could successfully pinpoint the defect location and scope.

**Table 5.** Comparison of different irradiance and filtering methods.

| Irradiance W/m$^2$ | Filter and Methods | | | |
| --- | --- | --- | --- | --- |
| | **Mean Filter** | **Median Filter** | **Improve Box Filter** | **Improve Box Filter and 3D Image** |
| 700 W/m$^2$ | O | O | O | V |
| 500 W/m$^2$ | X | O | O | V |
| 850 W/m$^2$ | O | O | O | V |

V: easy to identify; O: not easy to identify; X: unrecognizable.

When the irradiance was 500 W/m$^2$, the mean filter could not define the defect position clearly, while the median and box filters could also not easily pinpoint the defect location. It was found that when the box filter was converted into a 3D image, the defect area could then be easily located. Similarly, when the irradiance was 850 W/m$^2$, the mean and median filtering could not easily pinpoint the defect location, but when the box filter was converted into a 3D image, the defect area could then be located, as shown in the green box in Figure 9e. The above findings demonstrate that our innovative method of improved box filtering and the establishment of 3D images could be effectively applied in the analysis of solar modules.

Furthermore, the analyzed images could generate temperature information using different colors. By matching with the color bar on the right-hand side of the image (color bar's temperature is in °C), the temperature value could be known. Finally, the temperature difference between normal and defective cells was obtained, and the defect position could be successfully located. The innovative method of this study uses improved box filtering for analysis. To sum up, we cannot simply take the color and temperature of the IR image as the indicators for judging defects. The method of filtering after grayscale is suitable for use as a qualitative indicator, as it determines the hot-spot zone by the degree of whiteness. Through this research, it was found that the grayscale image could not easily define the scope and extent of the defect (i.e., the temperature in the white area), but 3D images could fulfill that purpose.

The 3D image can not only illustrate the defect zone, but also is able to provide the temperature value of the defect zone, and can indicate the scale of defects through the cumulative temperature chart. Hence, 3D images are qualified as a quantitative and qualitative indicator.

## 5. Conclusions

This research used a drone equipped with an IR camera and instant image transmission function, as well as utilizing the MATLAB image analysis method to analyze the IR images. This methodology can generate an instant IR image for monitoring the health conditions of solar modules. In particular, the relative temperature of the detected solar module can be read in the 3D image. By doing so, the problem of unrecognizable IR images is overcome, and the best solution is identified.

The goal of this innovative method is to use an infrared imaging system with drone functionality to detect faults in solar power generation systems, and to analyze the health status of the module based on the box-filtered IR image obtained from the drone. The results prove that this method is more accurate than median and mean filtering, and is able to pinpoint the defect location, while the health and the defect scope of the cell can

also be judged. The most significant contribution of this study is to show that the IR image with irradiance of 500 W/m² can be analyzed by means of a 3D image processed with improved box filtering. Utilizing this method, the defect location can be precisely identified, and monitoring can be conducted in wider time zones even during cloudy days, or when irradiance is just around 500 W/m².

Together with this research, four image analysis steps are proposed: In the first step, the original IR image is filtered and converted into a grayscale image. Secondly, the image is transformed into a 3D image, and the temperature value can be obtained through matching the temperature color with the relative color bar. This innovative method can not only significantly shorten the inspection time, but also can analyze the defect location of the IR image immediately. This is particularly convenient for maintenance personnel, as they can locate the defective area and simultaneously deduce the quantity of the faulty cells.

These research experiments were conducted in solar photovoltaic farms and solar modules installed on the rooftops of high-rise buildings. It was found that these monitoring procedures are also applicable to small solar photovoltaic installations in metropolitan areas, thereby facilitating the construction of diversified solar farms in crowded areas.

Looking forward, the next phase of this research will be continued. More information about solar module images with different defects will be collected to create a database for future in-depth investigation and development of learning-based methods to identify solar panel defects. It is hoped that by combining the image analysis technology of this study with AI technology, the ability of automated identification can be further strengthened, so as to save inspection manpower and improve maintenance efficiency in a significant way.

**Author Contributions:** K.-C.L. devised the experimental strategy and carried out this experiment, and also wrote the manuscript and contributed to the revisions. H.-Y.W. and H.-T.W. contributed to writing—original draft preparation. All authors have read and agreed to the published version of the manuscript.

**Funding:** This research received no external funding.

**Data Availability Statement:** Data are contained within the article.

**Conflicts of Interest:** The authors declare no conflict of interest.

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
