# Peer review of "Using Drones for Thermal Imaging Photography and Building 3D Images to Analyze the Defects of Solar Modules"

_inventions, doi:10.3390/inventions7030067_

Round 1

Reviewer 1 Report

The authors have used the infrared imaging system with drone to detect faults in the solar power generation system. 

The approach of use an infrared  imaging system with UAV functionality to detect defects is not novel but the application of this methodology in this case “to detect  faults in solar power systems” is interesting.

 The english language used in the article appears good . The images and tables are well-explicative.

The introduction section needs to be extend for other methodologies , I suggest to read and cite the article previously published:

Sciuto, G. L., Napoli, C., Capizzi, G., & Shikler, R. (2019). Organic solar cells defects detection by means of an elliptical basis neural network and a new feature extraction technique. Optik, 194, 163038.

Lo Sciuto, G., Capizzi, G., Shikler, R., & Napoli, C. (2021). Organic solar cells defects classification by using a new feature extraction algorithm and an EBNN with an innovative pruning algorithm. International Journal of Intelligent Systems, 36(6), 2443-2464.

Reviewer 2 Report

The paper describes techniques to detect defects in solar modules using thermal imaging. Mean, median, box, and 3D RGB box filter approaches are used to achieve the goal. The paper shows the results for different radiation levels and highlights the utility of the 3D RGB box filter.

There are two significant results that the authors need to merit publication.

1. Comparison with existing systems is missing. Why would one use this technique in place of others, needs to come out both in the Introduction as an explanation and quantitative results in the evaluation.

2. Table 5 needs to have quantitative results instead of the qualitative ones provided by the authors. It needs to be clarified what the authors define as easy to detect or hard to detect.

Reviewer 3 Report

The article describes a new method of identifying damage to solar panels using a drone.

Unfortunately, the description of the research methodology includes some shortcomings:

1. Descriptions of experiments confirming the proposed method are not developed correctly. Chapter 3 partially describes the experiment with the method and the drone, but this chapter also contains references to the bibliography that should be included in the description of the state of the art. The confusion of the descriptions of the state of the art and the experiment causes chaos in the article.

2. Chapter 4, which should contain the result, contains a description of another experiment. The lack of separation of the descriptions of the methodology of the experiments and results, as well as the justification and comparison of the results of the methodology with other methods, are incomplete and mixed up.

3. There is no certainty that the experiment with the damaged surface of the panel described in chapter 4 allows for the identification of damage to individual cells, the detected changes may well result from the physical change of the surface itself and not the work of individual cells. The experiment in this form is not complete.

4. There is no methodical description of the experiments and no statistical justification resulting from the conducted experiments, the proposed method of evaluating photovoltaic panels and a critical comparison to other evaluation methods for these experiments.

5. The results of the experiments are not supported by detailed electrical measurements allowing to identify cell damage of photovoltaic panels.

Round 2

Reviewer 2 Report

There have not been sufficient updates from the authors' end to answer my queries. Comparison with existing systems is still missing in the revised version. The authors have compared the three approaches provided in the paper, but there is no quantitative comparison with systems already referred to in the Introduction. 

Similarly, quantitative values are not used in Table 5. It is not clear how the authors quantify easy-to-detect or hard-to-detect.

Reviewer 3 Report

The authors did not address the comments of the reviewer. The method is based on a single experiment, comparing 2 panels, one without defects and the other with artificially produced defects, raises serious doubts. Concluding on a single experiment does not take into account the influence of other factors already visible in this experiment, e.g. panel fixing. The lack of an analysis based on data from a statistically justified group of cases is a serious methodological error and authorizes the authors only to make a thesis and not to prove the correctness of the method. The authors completely ignored these comments from the reviewer. Additionally, the experiment shows a strange convergence of the general temperature distribution on both panels, probably resulting from the construction or mounting of the panels, which has a significant impact on the level of results. The lack of a broader sample of the set of the results of the experiment with different mounting methods, wind direction,  and various configurations and layouts of bases makes it difficult or even impossible to correctly classify of malfunctions.